# Molecular Targets of Novel Therapeutics for Diabetic Kidney Disease: A New Era of Nephroprotection

**DOI:** 10.3390/ijms25073969

**Published:** 2024-04-03

**Authors:** Alessio Mazzieri, Francesca Porcellati, Francesca Timio, Gianpaolo Reboldi

**Affiliations:** 1Diabetes Clinic, Department of Medicine and Surgery, University of Perugia, 06132 Perugia, Italy; alessiomazzieri90@gmail.com (A.M.), francesca.porcellati@unipg.it (F.P.); 2Division of Nephrology, Department of Medicine and Surgery, University of Perugia, 06132 Perugia, Italy; francesca.timio@ospedale.perugia.it

**Keywords:** diabetic kidney disease, pathogenetic mechanisms, RAAS blockers, SGLT2 inhibitors, mineralocorticoid receptor antagonists, glucagon-like peptide-1 receptor agonists, endothelin receptor antagonists, aldosterone synthase inhibitors

## Abstract

Diabetic kidney disease (DKD) is a chronic microvascular complication in patients with diabetes mellitus (DM) and the leading cause of end-stage kidney disease (ESKD). Although glomerulosclerosis, tubular injury and interstitial fibrosis are typical damages of DKD, the interplay of different processes (metabolic factors, oxidative stress, inflammatory pathway, fibrotic signaling, and hemodynamic mechanisms) appears to drive the onset and progression of DKD. A growing understanding of the pathogenetic mechanisms, and the development of new therapeutics, is opening the way for a new era of nephroprotection based on precision-medicine approaches. This review summarizes the therapeutic options linked to specific molecular mechanisms of DKD, including renin-angiotensin-aldosterone system blockers, SGLT2 inhibitors, mineralocorticoid receptor antagonists, glucagon-like peptide-1 receptor agonists, endothelin receptor antagonists, and aldosterone synthase inhibitors. In a new era of nephroprotection, these drugs, as pillars of personalized medicine, can improve renal outcomes and enhance the quality of life for individuals with DKD.

## 1. Introduction

Diabetic kidney disease (DKD) is a chronic microvascular complication of diabetes mellitus (DM) and the leading cause of end-stage kidney disease (ESKD) [1]. The pathogenesis of DKD, influenced by genetic and environmental triggers, is complex, heterogenous, and sometimes obscure [2]. The specific molecular mechanisms, contributing to DKD in about one-third of patients with diabetes, are still unclear [3]. Although previous studies were historically focused on glomerular damage, there is increasing literature on the role of tubular injury in the pathogenesis of DKD [4]. Glomerular hyperfiltration and alterations of the extracellular matrix lead to mesangial matrix expansion and increased thickness of the glomerular basement membrane (GBM) [3]. The degree of podocyte damage and glomerulosclerosis correlates closely with albuminuria and the decline of renal function [3]. However, the rate of kidney function loss is more closely related to tubular damage and interstitial fibrosis than to glomerular damage [4]. Apart from this distinction between the two sites of renal damage, the onset of DKD involves different pathways, which are broadly classifiable into metabolic, oxidative, inflammatory, fibrotic, and hemodynamic [5]. The complex interplay of different processes appears to drive disease progression, although the individual contribution of each process varies. In over 30 years, several drugs potentially targeting the molecular basis of DKD have been developed [6]. Beyond the renin–angiotensin–aldosterone system (RAAS) blockers, recent evidence from clinical trials established sodium-glucose cotransporter 2 inhibitors (SGLT2i), non-steroidal mineralocorticoid receptor antagonists (NS-MRA) and glucagon-like peptide 1 receptor agonists (GLP-1 RAs) as effective add-on therapies for DKD [7]. Even though it is still hold that RAAS blockers should be started and titrated to the maximally tolerated dose before adding any new drug [8], mounting clinical evidence has determined a paradigm shift supporting a pillared approach to reduce kidney outcomes. The concept of pillars of therapy, successfully applied in heart failure, suggests that all agents of more superior efficacy on kidney outcomes than a single drug class alone should be used in combination [8]. Extending the number of available pillars might have a positive impact on residual risk. Emerging evidence suggests that both endothelin-A receptor antagonists (ETA-RAs) and aldosterone synthase inhibitors (ASI) appear promising pillars [9,10].

Modern strategies based on agents with complementary mechanisms of action pave the way for a new era of nephroprotection based on personalized medicine approaches [11]. Major guidelines already recommend a multi-agent strategy, personalized to the individual residual renal and/or cardiorenal risk [12,13]. 

In this narrative review, we will summarize the molecular mechanisms, their interconnections, and their links to the development of novel therapeutic options and strategies for DKD.

## 2. Molecular Mechanisms of DKD

### 2.1. Metabolic Factors

Chronic hyperglycemia plays a major role in the pathogenesis of DKD [14]. One important function of the kidney is the reabsorption of glucose from glomerular filtrate. Under normal conditions, more than 99% of the total filtered glucose (around 180 g/day) is reabsorbed into the circulation by the sodium-glucose cotransporters (SGLTs) family in the proximal convoluted tubule [15]. Specifically, SGLT2 is responsible for reabsorption of more than 90% of the filtered glucose in the kidney [16]. In case of DM, the renal threshold is increased and expression of SGLT2 gene is upregulated, leading to increased glucose reabsorption from glomerular filtrate and a subsequent enhancement of blood glucose levels [14]. Beyond the systemic consequences, the worsening of hyperglycemia causes directly damages in the kidney, including: increased expression of transforming growth factor-β (TGF-β) and overproduction of the mesangial matrix; production of ROS and advanced glycation end products (AGEs) [14]. Moreover, hyperglycemia is related to the mechanism of renal hypoxia implicated in the onset and worsening of DKD [17]. Hypoxia-inducible factor 1 (HIF-1), a basic helix-loop-helix transcription factor that consists of an oxygen-sensitive α-subunit and a constitutively expressed β-subunit, is a pivotal mediator of oxygen homeostasis [18]. Glucose is known to elicit the expression of HIF-1α in the mesangium, which eventually converges to transactivate downstream fibrotic factors, such as connective tissue growth factor (CTGF) and plasminogen activator inhibitor 1 (PAI-1), which are known to be involved in glomerular fibrosis [18]. At last, hyperglycemia is intrinsically linked to insulin resistance, which facilitates a dyslipidemia characterized by high plasma TG-rich lipoproteins, low-plasma high-density lipoprotein cholesterol (HDL-C), and increased concentrations of small-sized and modified low-density lipoprotein particles (ox-LDL) [14]. Several evidences have shown that increased lipid accumulation in tubular epithelial cells (TECs) lead to lipotoxicity, which contributes to mechanism of tubular fibrosis [19]. These results suggest the role of the crosstalk between lipid and glucose dysmetabolism in the pathogenesis of DKD.

### 2.2. Oxidative Stress

Oxidative stress, defined as an imbalance between the formation of highly reactive molecules and the antioxidant mechanism, has long been recognized as a crucial factor in the pathogenesis of DKD [20]. Reactive oxygen species (ROS) are chemicals molecules generated in cells, such as superoxide anion (O_2_^−^), hydroxyl radical (OH), and hydrogen peroxide (H_2_O_2_) [20]. Among them, Nicotinamide adenine dinucleotide phosphate (NADPH) oxidase is the most important source of superoxide anion and is upregulated in the kidney of diabetic people. In addition to its direct effects on cellular protein and DNA [21], ROS activates several signaling cascades such as the transcriptional factor NF-κB that is implicated in the inflammatory damage of DKD [22]. Another important pathway whereby glucose can activate oxidative stress is the AGE signaling. AGEs are generated exogenously and endogenously through non-enzymatic reactions of amino acids, proteins, and glucose [23]. Specifically, the flux of glucose through the polyol pathway increases the formation of toxic AGE compounds, leading to the production of free radicals. Accumulated evidence shows that increased ROS accelerates protein glycation in the body with consequent glycative stress [20]. For instance, glycated albumin contributes to kidney fibrosis in DKD via upregulation of myostatin, a family member of the TGF-β superfamily [24]. Given that the endoplasmic reticulum (ER) is a crucial organelle in the maintenance of protein quality, the link between glycation and ER function may be critical to the quality control of protein [24]. Indeed, ER dysfunction by hyperglycemia plays an important role in DKD, especially by high glucose-induced tubular damage. Specifically, AGEs can cause oxidative stress and fibrosis in an indirect manner through the binding with the Receptor of AGEs (RAGE), a multi-ligand cell surface receptor that can be induced under diabetic conditions [20]. Indeed, the intracellular accumulation of AGEs stimulates various signaling pathways, including the inflammatory response via regulation of NF-κB [20], as well as ROS signaling cascades. Evidence on animal models shows that AGEs cause pathological changes in DKD, such as glomerular hypertrophy, mesangial expansion, glomerular basement membrane thickening, and glomerular sclerosis [25]. Hence, both elevated ROS and AGEs form a causal link between hyperglycemia and DKD progression.

### 2.3. Inflammatory Pathway

The production of ROS increases via the transcription factor NF-κB, the expression of cell adhesion molecules on the surface of renal endothelial cells [26]. These molecules recruit leukocytes to the site of inflammation and promote the adhesion of these cells to endothelium and mediate subsequent transmigration [27]. Among adhesion molecules, intercellular adhesion molecule-1 (ICAM-1) and vascular adhesion molecule-1 (VCAM-1) are associated with the development and progression of DKD [26]. Specifically, the recruitment of monocytes and macrophages into the kidney is a key step in the pathophysiology of DKD [28]. The recruitment of macrophages in kidney tissues is directly correlated with the expression of inflammatory mediators, glomerular and tubular damage, interstitial fibrosis, and ultimately, decreased renal function [29]. Several cytokines and chemokines such as TNF-α, IL-1, IL-6, IL-18, CCL2, and CCL5 have been found to be upregulated in the kidney and are associated with enhanced vascular endothelial permeability, increased mesangial cell proliferation, extracellular matrix expansion, and increased albuminuria [26]. Studies also suggested that TNF-α triggers apoptosis and necrosis pathways; destabilization of the intraglomerular hemodynamics, thereby reducing glomerular filtration rate; and induction of NADPH-mediated cellular oxidative stress [26]. Pro-fibrotic mediators released by macrophages determine extracellular matrix deposition with consequent fibrosis and impaired function [30]. Indeed, macrophage accumulation in the kidney strongly correlates with serum creatinine levels, interstitial myofibroblast accumulation, and interstitial fibrosis scores [29]. In the kidney, the patrolling of monocytes orchestrates immune cell responses at the glomerular vascular interface, including recruitment and activation of neutrophils [31]. Moreover, resident macrophages and dendritic cells in the tubulointerstitium also contribute to disease progression by recruiting and activating lymphocytes [27]. In the context of adaptive immunity, T cells infiltrate the kidneys in DKD, while evidence for the involvement of B cells is limited [27]. Animal DKD models show an early infiltration by helper CD4 T cells, followed by a wave of macrophages and cytotoxic CD8 T cells, suggesting a Th1-mediated response [27]. Specifically, a T-cell homing into the kidneys is mediated by ICAM-1 expressed on renal endothelial, epithelial, and mesangial cells [27]. However, not all T-cell activity promotes DKD. Indeed, Tregs appear to be able to ameliorate DKD [27]. A study of diabetic nephropathy in db/db mice (Leptin-Receptor deficient mice) showed that Tregs could attenuate T2D-related kidney morphologic and functional abnormalities [32]. Furthermore, evidence from both animal models and human studies suggests that DKD is associated with a shift toward Th1/Th17 phenotypes and reduced Treg activity [27]. For these reasons, the innate and adaptive immunity are both fundamental in the pathogenesis of DKD.

### 2.4. Fibrotic Signaling

The tubulointerstitial fibrosis of DKD is associated with tubular atrophy and extracellular matrix accumulation [33]. This pathologic process is related to various fibrotic factors such as TGF-β, ET-1, fibronectin, collagen-1, and CTGF [5]. Among them, TGF-β has the pivotal role in the onset and progression of damage in DKD [34]. In 2021, Yang et al. showed that TGF-β downstream Suppressor of Mothers Against Decapentaplegic (SMAD) molecules promote autophagy dysregulation in tubular epithelial cells [35], while Hong et al. demonstrated the proangiogenic effects of TGF-β1 signaling in glomerular endothelial cells [36]. Apart from TGF-β, ET-1 also has a prominent role in the pathogenesis of DKD. ET-1 is a potent vasoconstrictor that promotes efferent arteriolar vasoconstriction, glomerular hypertension, and sodium retention along with activating proinflammatory and profibrotic pathways [37]. Plasma levels of ET-1 are increased in response to hyperglycemia, dyslipidemia, endothelial dysfunction, and oxidative stress [38]. In addition, increased renal ET expression is associated with mesangial proliferation and podocyte injury [39]. Among the other fibrotic factors, the accumulation of fibronectin in the glomerular mesangium is related to deterioration of kidney function in DKD [40]. Moreover, experimental evidence on animal models showed also that collagen-1 is involved in the pathogenesis of renal fibrosis in DKD [5]. The role of CTGF expression levels correlates with the degree of glomerulosclerosis and tubulointerstitial fibrosis. CTGF concentrations in diabetic patients seems correlated with higher urinary albumin excretion and lower estimated glomerular filtration rate (eGFR) [41].

### 2.5. Hemodynamic Mechanisms 

Glomerular hyperfiltration is frequent in early DKD, and glomerular enlargement is a key characteristic [42]. Kidney enlargement is often seen in DKD, and its mechanism is related to elongation of the proximal tubules caused by an increase in single nephron glomerular filtration rate (snGFR) and glomerular hypertension [5]. In diabetic nephropathy, glomerular hyperfiltration is the first stage in the pathogenesis, leading to progressive albuminuria, declining GFR, and finally ESKD, although the mechanism is not yet fully understood [43]. Glomerular hypertension is also influenced by lifestyle factors, such as diet and body weight [5]. Hyperglycemia interacts with elevated levels of circulating amino acids caused by a high-protein diet and becomes a trigger for glomerular hyperfiltration [44]. However, the activation of RAAS is the pivotal hemodynamic mechanism involved in hyperfiltration and progression of DKD [45]. Angiotensin II is one of the bioactive members of the RAAS and has been confirmed to be elevated in DKD [46]. Angiotensin II has various physiologic effects and is known to promote ROS production [47]. In 2018, Ilatovskaya et al. reported that podocyte damage through angiotensin II-mediated calcium influx into podocytes contributes to renal injury in DKD [48]. Aldosterone also plays an important role in the pathophysiology of DKD [49]. Ritz et al. reported that aldosterone upregulates unfavorable growth factors such as PAI-1 and TGF-β, which promote macrophage infiltration and consequent renal fibrosis [50]. Indeed, aldosterone binds to mineralocorticoid receptors (MRs) and this binding leads to recruitment of transcriptional cofactors and transcription of genetic target, including proinflammatory and profibrotic genes [51]. Although it was well-known that RAAS activation is deeply involved in the pathogenesis of DKD in terms of the glomerular component, how the presence of RAAS in the renal tubules affects the pathogenesis of DKD remains not fully clear [5]. Along with the RAAS, the system related to SGLT2 is another mechanism related to glomerular hyperfiltration [5]. SGLT2 increases the reabsorption of glucose in the proximal tubules, thereby reducing the delivery of sodium chloride to the macula densa [44]. As a result, tubulo-glomerular feedback is reduced, afferent arterioles are dilated, and angiotensin II is increased in efferent arterioles, resulting in vasoconstriction [52]. These effects increase glomerular perfusion and intraglomerular pressure, leading to glomerular hyperfiltration and progression of DKD. 

## 3. Therapeutic Options for DKD

The interplay of different processes (metabolic factors, oxidative stress, inflammatory pathway, fibrotic signaling, and hemodynamic mechanisms) are therefore the drivers to the onset and progression of DKD (Figure 1).

Based on the molecular mechanisms of DKD, several therapeutic drugs have been developed for clinical use (Table 1). A deeper understanding of the molecular mechanisms of DKD onset and progression have led to development of precision medicine approaches to the prevention of renal disfunction and treatment. These therapeutic options include renin–angiotensin–aldosterone system blockers, mineralocorticoid receptor antagonists, SGLT2 inhibitors, glucagon-like peptide-1 receptor agonists, and endothelin receptor antagonists (Figure 2).

### 3.1. RAAS Blockers

Since the past few decades, blockage of RAAS inhibition has been the most pillar for the management of DKD [45]. The Angiotensin-converting-enzyme inhibitors (ACEi) was the first class of drugs targeting the RAAS. Several preclinical and clinical studies have reported renoprotective effects of ACEi [45]. By limiting the Angiotensin II production, ACEi treatment reduces glomerular hypertension and glomerular permeability to urinary albumin resulting in reduced proteinuria [67]. In the “Collaborative Study Group Captopril trial” (CSG Captopril trial), captopril treatment to type 1 DM patients with DKD reduced the risk of a doubling of serum creatinine and DKD-associated death, dialysis, or transplantation almost by half (50%) in comparison to the placebo, during a median of 3-year follow-up [53]. This study paved the way for the subsequent trials with Angiotensin receptor blockers (ARBs). In the “Reduction of Endpoints in Non-insulin-dependent DM with the Angiotensin II Antagonist Losartan” (RENAAL) study, losartan abridged the occurrence of a doubling of the serum creatinine concentration by 25% [54]. In the “Irbesartan Diabetic Nephropathy Trial” (IDNT) irbesartan, when compared with the placebo, had a lower relative risk for the primary endpoint (doubling creatinine, development of ESKD, and death), especially for the doubling of creatinine [55]. In the “Diabetics Exposed to Telmisartan And enalaprIL” (DETAIL) study, telmisartan improved the decline in GFR over 5 years in type 2 DM patients with early nephropathy [56]. In “A comparison of telmisartan versus losartan in hypertensive type 2 diabetic patients with overt nephropathy” (AMADEO) study, telmisartan produced a 29.8% reduction in urine albumin–creatinine ratio, whereas that with losartan was only 21.4% [57]. Additionally, the “ONgoing Telmisartan Alone and in combination with Ramipril Global Endpoint Trial” (ONTARGET) [68] demonstrated that telmisartan, as compared with ramipril, had a similar relative risk of kidney impairment, whereas the combination therapy had a significant increase in its relative risk. Collectively, both ACEi and ARBs, but not their combination, are fundamental elements of RAAS modulation in DKD management. 

Upstream inhibitors of RAAS system can modulate angiotensin II and aldosterone, with potentially favorable hemodynamic and structural effects on the kidney [45]. Among direct renin inhibitors (DRIs), aliskiren became the first clinically approved compound in 2007. However, it didn’t seem to be more effective or better tolerated than existing treatments [69]. Other approaches, targeting the RAAS upstream, involve the small interfering RNA (siRNA) that function via the RNA interference pathway and inhibit angiotensinogen RNA translation [70]. In this context, Zilebesiran is a siRNA that binds to the hepatic asialoglycoprotein receptor, resulting in decreased hepatic production of angiotensinogen (AGT) [69]. The suppression Zilebesiran-mediated of hepatic but not renal AGT prevents renal Angiotensin II generation, avoiding the consequences of total RAS blockade [70]. Recent clinical trials [71,72] showed that zilebesiran is an effective antihypertensive agent that may also have beneficial effects in other conditions associated with activation of the RAAS, such as kidney disease [73].

### 3.2. Mineralcorticoid Receptor Antagonists

Both steroidal and nonsteroidal mineralocorticoid receptor antagonists (MRAs) can cause on MRs a different conformational change that inhibits aldosterone from binding and its consequent downstream [51]. Steroidal MRAs interact with cofactors that affect gene transcription; hence, they function as partial MR agonists [74]. Spironolactone and eplerenone are steroidal MRAs that showed therapeutic benefits in cardiovascular disease (CVD), especially heart failure, and improved proteinuria in patients with DKD in a clinical cohort study [51]. However, spironolactone and eplerenone also cause anti-adrenergic side effects, hyperkalemia, and impairment of kidney function in patients with chronic kidney disease (CKD). Therefore, the practical use of these compounds is often limited by unwanted side effects [51]. Finerenone is a nonsteroidal MRA, which suppresses MR signaling at multiple levels. Finerenone inhibits cofactor recruitment to the MR from the cytoplasm to the nucleus and additionally, the gene regulation profile of finerenone differs from that of steroidal MRAs [75]. The antifibrotic activity and the specific anti-inflammatory effect of finerenone are more potent than steroidal MRAs [51]. The nephroprotective effect of finerenone in DKD is in fact driven by its anti-inflammatory, antifibrotic, and antioxidative properties [76]. A phase II “Mineralocorticoid Receptor Antagonist Tolerability Study” (ARTS) program showed that finerenone was effective at least as spironolactone in decreasing biomarkers for hemodynamic stress, while inducing less hyperkalemia and a decrease in renal function [77]. In 2021, finerenone was approval by the Food and Drug Administration (FDA) to reduce the risk of kidney function decline, kidney failure, cardiovascular death, non-fatal heart attacks, and hospitalization for heart failure in adults with chronic kidney disease associated with type 2 diabetes. The decision was based on the findings of the “Finerenone in Reducing Kidney Failure and Disease Progression in Diabetic Kidney Disease” (FIDELIO-DKD) trial [58], and the “Finerenone in Reducing Cardiovascular Mortality and Morbidity in Diabetic Kidney Disease” (FIGARO-DKD) trial [78]. The “FInerenone in chronic kiDney diseasE and type 2 diabetes: Combined FIDELIO-DKD and FIGARO-DKD Trial programme analysis” (FIDELITY) pools these complementary studies with similar designs, assessments, and conduct [79]. Taken together, these studies demonstrated the cardiovascular and renal benefits of finerenone in patients with type 2 DM and albuminuria who were on standardized treatment (maximum tolerated dose of RAAS inhibitor, glycemic control, and blood pressure management) [51,79]. Specifically, finerenone showed a significant risk reduction compared with placebo in all composite cardiovascular (cardiovascular death, nonfatal myocardial infarction, nonfatal stroke, or hospitalization for heart failure) and renal outcomes (a sustained ≥ 57% decrease in eGFR from baseline over ≥4 weeks or renal death) [51,79]. Interestingly, the blood pressure changes in the participants in the trials were modest, which indicates that the cardiovascular and renoprotective effects of finerenone cannot be fully explained by changes in blood pressure [79]. The inclusion of albuminuric patients might somewhat limit its generalizability to non-albuminuric patients; thus, further studies are necessary [51]. 

Beyond MRAs, the selective aldosterone synthase inhibitors (ASI) is emerging as a new class of drugs, specifically targeting aldosterone synthesis upstream of the MR [80]. MR is not only activated by aldosterone, but also by cortisol, and by not specific stimuli such as hyperglycemia, high salt intake, and oxidative stress. Baxdrostat [81] and lorundrostat [82] effectively lowered blood pressure in patients with resistant or uncontrolled hypertension, but their effects on renal outcomes have not been not investigated so far [83]. Very recently, the efficacy of BI 690517, a highly selective ASI, was tested in CKD patients. In a phase II placebo-controlled renal outcome trial, 586 CKD patients were randomized to BI 690517 [66] either as a monotherapy or in combination with empagliflozin. The experimental drug, BI 690517, showed a dose-dependent reduction of albuminuria, and when used on top of empagliflozin exhibited an additive efficacy. The incidence of hyperkalemia was higher with BI 690517 compared to placebo, although the median increase in serum potassium was lower across all dose groups when BI 690517 was administered along with empagliflozin.

### 3.3. SGLT2 Inhibitors

In 2014, the introduction of SGLT2 inhibitors (SGLT2i) created a renewed excitement in the improvement of DKD therapy [8]. Although they were originally designed to lower glucose by promoting urinary glucose excretion, it was then appreciated that SGLT2i were, in fact, cardiorenal risk-reducing drugs [84]. Initially, the nephroprotective benefits of SGLT2i were gleaned from secondary data analyses of trials with time to major adverse cardiovascular events as a primary outcome [8]. The “Canagliflozin and Renal Events in Diabetes with Established Nephropathy Clinical Evaluation” (CREDENCE) [59] study was the first nephropathy outcome trial to specify its primary outcome as ESKD, doubling of creatinine level, or death from renal or cardiovascular causes with an SGLT2i. Canagliflozin achieved a 30% lower relative risk of reaching the primary end point and a 32% lower relative risk of ESKD progression. Similar to findings in the CREDENCE trial, the “Dapagliflozin and Prevention of Adverse Outcomes in Chronic Kidney Disease (DAPA-CKD) trial showed that dapagliflozin achieved a 44% reduction in relative risk of sustained 50% reduction in eGFR, ESKD, or death from renal or cardiovascular causes [60]. Most recently, “The Study of Heart and Kidney Protection With Empagliflozin” (EMPA-KIDNEY) [61] showed that empagliflozin reduced the risk of ESKD progression or cardiovascular death by 28% compared with placebo. Interestingly, in this trial there were people with both normal and increased albuminuria, as well as patients with and without DM. Accordingly, the Kidney Disease: Improving Global Outcomes (KDIGO) and American Diabetes Association (ADA) guidelines recommended the use of SGLT2i in patients with DKD and cardiovascular disease on top of the standard of care [85]. Notably, when SGLT2i were given on top of ACEI/ARB, the beneficial effect on kidney function decline was some 30–40% greater than patients receiving ACEI/ARB alone [8]. Beyond the well-known favorable action of SGLT2i on tubulo-glomerular feedback and glomerular hyperfiltration, they also exhibit pleiotropic effects. Several studies in humans and animal diabetic models showed that SGLT2i reduce inflammation, extracellular matrix turnover, and fibrosis [85]. 

### 3.4. Glucagon-like Peptide-1 Receptor Agonists

GLP-1 is a peptide hormone produced by the gut epithelium that regulates blood glucose levels through the activation of the GLP1-R in the pancreas to increase insulin and decrease glucagon secretion [86]. GLP-1 RAs are currently approved for type 2 diabetes and obesity [86]. However, in randomized trials, GLP-1 RA showed significant kidney-protective effects largely driven by albuminuria reduction [7]. The protective effect of GLP-1RA on the kidney seems largely independent from glycemic control, and likely related to the inflammatory and pro-fibrotic signaling modulation [87]. The anti-inflammatory potential of GLP-1 RAs depends on both their direct effect on immune cells [88] and indirect effects due to weight loss. GLP-1 and its cleavage products were nephroprotective in murine models of diabetic nephropathy through the reduction of renal infiltration by inflammatory cells [89]. Specifically, experimental evidence showed that GLP-1 RAs can reduce the inflammatory damage by modulating RAGE signaling. Liraglutide in diabetic mice remodeled a network of nutrient synthesis and transport, and promoted redox sensing signals in proximal tubular cells, podocytes, and macrophages [90]. Additionally, in the proximal tubule, the activation of GLP1-R leads to the onset of the cAMP signaling cascade with phosphorylation of Na^+^/H^+^ exchanger isotope 3 (NHE3) and reduction of its function. Hence, GLP1-R signaling can contribute to the regulation of sodium balance and maintain blood pressure in the normal range [91]. 

The data for the potential kidney benefits of GLP-1 RA have been extrapolated from cardiovascular outcomes trials (CVOT) [92]. The “Liraglutide and Cardiovascular Outcomes in Type 2 Diabetes” (LEADER) study [93], which evaluated liraglutide, included 23% of patients with a history of CKD. The trial showed a significant 22% decrease in nephropathy events, driven by a major reduction of 26% in the development of macroalbuminuria. In the “Semaglutide and Cardiovascular Outcomes in Patients with Type 2 Diabetes” (SUSTAIN-6) trial [94], there was a 36% decrease in new or worsening nephropathy with the injectable GLP-1 receptor agonist semaglutide. This was primarily driven by a 46% decrease in development of persistent macroalbuminuria with no effect on doubling of serum creatinine. Similarly to effects of liraglutide and semaglutide, dulaglutide showed kidney benefits in two trials. The “Dulaglutide versus insulin glargine in patients with type 2 diabetes and moderate-to-severe chronic kidney disease” (AWARD-7) trial [95] enrolled adult patients with type 2 diabetes and CKD stages 3–4, in treatment with ACEi/ARBs and found that dulaglutide had significantly less eGFR decline to insulin glargine (mean −0.7 vs. −3.3 mL/min/1.73 m^2^, respectively). Additionally, “The Researching Cardiovascular Events with a Weekly Incretin in Diabetes” (REWIND) trial [96] denoted a 23% lower risk of development of macroalbuminuria and a significant 15% risk reduction in the renal composite endpoint compared to placebo. Finally, in the “Cardiovascular and Renal Outcomes with Efpeglenatide in Type 2 Diabetes” (AMPLITUDE-O) trial [97], efpeglenatide showed that kidney outcomes were reduced by 32% compared to the placebo group. Although 15.2% of patients in the study were received SGLT2i at study entry, the kidney protection of efpeglenatide was independent from these drugs. However, to date there no published GLP-1 RA dedicated primary renal outcome trial in patients with CKD. The final result is of the FLOW trial “Effect of Semaglutide Versus Placebo on the Progression of Renal Impairment in Subjects With Type 2 Diabetes and Chronic Kidney Disease” are eagerly awaited [62]. The FLOW study aimed to assess whether the once-weekly injection of semaglutide could delay the progression of kidney disease and reduce the risk of death related to the kidneys or the cardiovascular system in patients with type 2 diabetes and CKD. The study was originally designed to recruit at least 3500 patients followed for up to 5 years, but on 10 October 2023 the trial was stopped early for efficacy. The main findings of the FLOW trial will be released and published in the first half of 2024 [98]. 

Beyond GLP1 RA, there is growing interest regarding dual gastric inhibitory polypeptide (GIP) and GLP-1 receptor agonist, tirzepatide. Secondary analyses of the tirzepatide trials have shown significant benefits on kidney outcomes [92]. In a post-hoc analysis of SURPASS-4 [99] tirzepatide compared with insulin glargine showed a remarkable 42% (*p* = 0.0008) relative risk reduction of the kidney composite outcome including time to first occurrence of eGFR decline of at least 40% from baseline, end-stage kidney disease, death owing to kidney failure, or new-onset macroalbuminuria. The overall risk reduction was largely driven by the effect of tirzepatide on albuminuria, in keeping with other studies with GLP-1 RAs [100]. Interestingly, a slower rate of eGFR decline was found in the 25% of participants already treated with SGLT2i [99,100]. These findings might be linked to GIP agonism, which might exert an additional protective effect on the kidney [100].

The renal effects of cotadutide, a dual GLP-1 and glucagon receptor agonist, was recently investigated in a phase IIa trial [101] originally designed to test its effect on plasma glucose changes following a mixed-meal tolerance test. Notably, in a limited number of patients with baseline micro- or macroalbuminuria (*n* = 18), urinary albumin–creatinine ratios decreased by 51% at day 32 with cotadutide versus placebo (*p* = 0.0504). This finding might be in part explained by glucagon receptor activation in the kidney. Studies in animal models showed that deficiency of glucagon receptors in nephron tubules leads to inflammatory damage, fibrosis, water, and electrolyte imbalances [102]. On the contrary, an activation of glucagon receptors may reduce perirenal adiposity, improve renal blood flow and glomerular filtration, and may indirectly influence solute transport in the proximal renal tubules [103]. Finally, further evidence might emerge on the role of glucagon receptors activation from a phase IIb double-blind trial with retatrutide, a new triple agonist of GIP, GLP-1, and glucagon receptors [104]. The study is in progress and will investigate the effect of retatrutide on a primary renal outcome (change from baseline in GFR) in overweight or obese patients with CKD.

### 3.5. Endothelin Receptor Antagonists

An increased production of the potent vasoconstrictor ET-1 contributes to cardiorenal risk in patients with DKD, and therefore is an actual therapeutic target [105]. Based on rodent models of diabetes, ET-1 inhibition has kidney-protective effects through restoring the glomerular endothelium, preventing podocyte loss, and reducing glomerular hypertension and matrix expansion [38]. ET-1 signals through two receptors: endothelin A (ETA) and endothelin B (ETB) [105]. Selective ETA antagonists are attractive, since the ETB receptor works to clear circulating ET-1 via the lungs, whereas antagonizing the ETA receptor can lower blood pressure, reduce fibrosis, and decrease kidney inflammation [105]. ETA-RAs have also shown efficacy for reducing albuminuria and the risk of renal outcomes in patients with DKD, yet they are also associated with fluid retention. High doses of relatively unselective ETA receptor antagonists increase in fact the risk of heart failure in DKD patients [105]. In the “Avosentan on time to doubling of Serum Creatinine, ENd stage renal disease or Death” (ASCEND) trial [63], avosentan reduced albuminuria by 44% (25 mg/d dose) and 49% (50 mg/d dose) compared with placebo in DKD patients, when added to an ACEi or ARB treatment. However, this trial was terminated early because of an excess of congestive heart failure events. In the “Study Of diabetic Nephropathy with AtRasentan” (SONAR) trial [64], atrasentan significantly reduced the risk of DKD progression and kidney failure compared with placebo in patients with DKD. However, fluid retention, anemia, and hospitalization for heart failure were more frequent in the atrasentan group compared to placebo [7]. Fluid retention is a frequent and treatment-limiting side effect of ETA receptor antagonists that might be offset by the diuretic and natriuretic effect of SGLT2i [105]. In this context, the combination of zibotentan and dapagliflozin may represent a novel therapeutic option for DKD and CKD, owing to different but potentially additive effects [105]. The “Zibotentan and Dapagliflozin for the Treatment of Chronic Kidney Disease” (ZENITH-CKD) was the first phase II clinical trial to evaluate this combination in 525 patients with CKD (≈50% with DKD) [65,105]. Due to the careful dose selection of zibotentan added to the diuretic and natriuretic effects of concomitant SGLT2 blockade, combination treatment effectively lowered the risk of fluid retention [65]. The 25–30% additional reduction in albuminuria observed with zibotentan in combination with dapagliflozin provides evidence supporting their additive effect. The magnitude of the effect is clinically relevant and might translate into long-term beneficial effects on kidney outcomes [106].

## 4. Conclusions

DKD is a complex disease driven by a series of disrupted metabolic, hemodynamic, inflammatory, and fibrotic processes [5]. The pathogenetic mechanisms are triggered and propagated by hyperglycemia and high blood pressure. Downstream effects triggered by these disturbances contribute to oxidative stress and the subsequent release of proinflammatory and profibrotic mediators [7]. The complex crosstalk of underlying mechanisms drives DKD onset and progression. 

Growing knowledge of DKD pathophysiology and mounting clinical evidence has determined a paradigm shift supporting a pillared approach to reduce kidney and cardiovascular outcomes. Blockade of the renin–angiotensin system is still the first and fundamental element of the DKD treatment pillars [8]. SGLT2i and MRAs have reached a solid position among the pillars, while GLP-1 RAs might be soon included as the fourth pillar [107]. However, it is still unclear if all agents should be started in parallel at lower dosages and then followed by up-titration in subsequent steps. 

Within the pillar framework, emerging therapies such as ETA-RA, ASI, dual GLP-1/GIP or GLP-1/Glucagon receptor agonists, and triple GLP-1/GIP/Glucagon receptor agonists might be viewed as potential additional agents and perhaps future pillars. Preliminary evidence supporting combination therapies, namely ETA-RA + SGLT2i or ASI + SGLT2i, set the scene for larger phase III studies examining whether the favorable effects will effectively lower the risk of cardiorenal outcomes in patients with DKD. The safety and tolerability of these new combinations appear acceptable when given on top of standard-of-care treatments [65,66], but these reassuring findings have yet to be confirmed in large-scale clinical trials. Beyond GLP-1 RA alone, the dual GIP/GLP-1 receptor agonist tirzepatide remarkably slowed the progression of DKD in a post hoc analysis of the SURPASS-4 trial [99]. Further ongoing kidney disease-focused primary-outcome trials will directly investigate the beneficial effects of dual GLP-1/Glucagon receptor agonists [101] and triple agonists [104].

A deeper understanding of novel therapeutics sets the scene for a personalized medicine approach to DKD [108]. Such a strategy is bound to enable tailoring the choice of primary agents and their combinations to the individual patient [6]. Personalized medicine, through a deep dive into molecular profiles and clinical data of DKD patients, may improve clinical outcomes, control costs, enhance therapeutic selection, and increase medication adherence [109]. However, choosing the optimal medications for the right patient at the right time remains challenging. In the past decades, several studies have been conducted to reduce renal risk in patients with DKD by targeting specific pathways. However, these trials disclosed a large variation in drug response often associated with unsuccessful findings [110,111]. A personalized medicine approach to clinical trials, using molecular profiling and drug discovery for tailoring the right therapeutic strategy for the right person at the right time, might effectively overcome the challenge [111,112]. 

As research continues, new knowledge is gained, fostering further identification of innovative targets that may strongly reduce DKD burden. Ultimately, in a new era of nephroprotection, a comprehensive precision medicine approach, spanning from molecular targets to clinical trial design, can improve renal outcomes and enhance the quality of life of patients with DKD. 

## Figures and Tables

**Figure 1 ijms-25-03969-f001:**
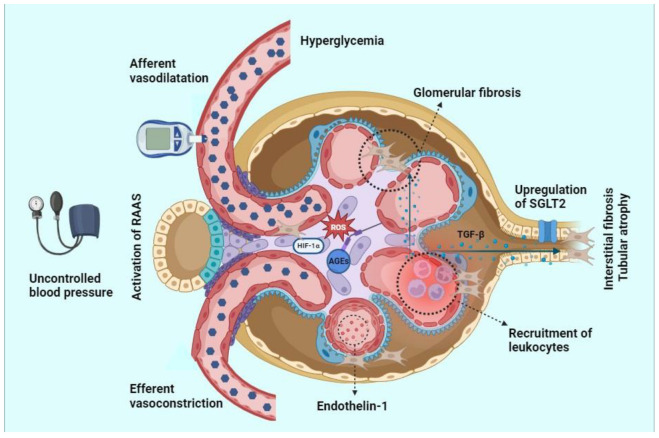
The molecular interplay that affects the onset and progression of glomerular and tubular damage in diabetic kidney disease (DKD). The blue hexagons represent glucose molecules and their number denote hyperglycemia measured by the glucometer. Uncontrolled blood pressure is symbolized by the sphygmomanometer. Red and light blue spheres indicate Endothelin-1 and TGF-β molecules. Solid arrows denote TGF-β profibrotic signaling. RAAS, renin–angiotensin–aldosterone system; HIF-1α, Hypoxia-inducible factor 1 alpha; ROS, reactive oxygen species; AGEs, advanced glycation end products; TGF-β, transforming growth factor-β; SGLT2, sodium-glucose cotransporter 2.

**Figure 2 ijms-25-03969-f002:**
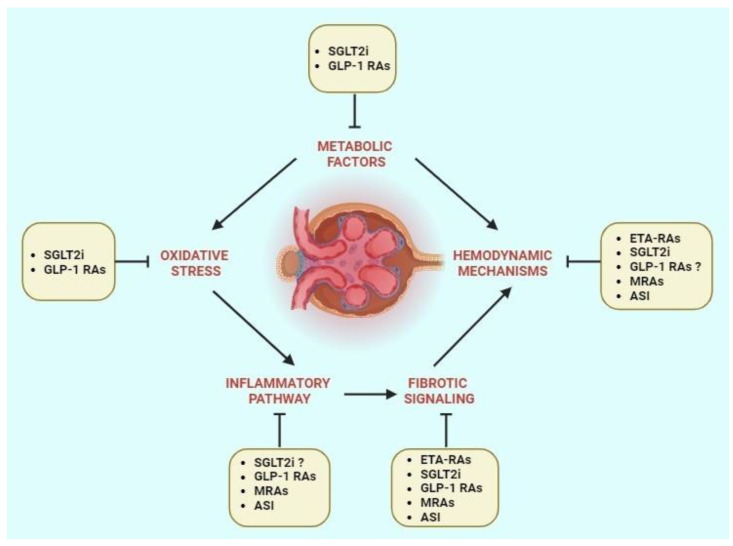
Therapeutic targets and new treatment options in DKD. SGLT2i, sodium-glucose cotransporter 2 inhibitor; GLP1 RAs, glucagon-like peptide-1 receptor agonists; MRAs, mineralocorticoid receptor antagonists; ASI, aldosterone synthase inhibitors; ETA-RAs, endothelin-A receptor antagonists. The symbol “?” denotes incomplete evidence supporting the specific role.

**Table 1 ijms-25-03969-t001:** Summary of the randomized controlled trials (RCT) mentioned in text with a kidney disease focused primary outcome. Angiotensin-converting-enzyme inhibitors (ACEi), Angiotensin receptor blockers (ARBs), non-steroidal mineralocorticoid receptor antagonists (NS-MRA), sodium-glucose cotransporter-2 inhibitors (SGLT2i), glucagon-like peptide-1 receptor agonist (GLP-1 RA), endothelin receptor antagonist (ETA RA), Aldosterone Synthase Inhibitor (ASI). ESKD, end-stage kidney disease; eGFR, estimated glomerular filtration rate; UACR, urine albumin–creatinine ratio.

Study, Year	Experimental Drug	Class	Study Population	Kidney Outcome	Reference
CSG Captopril, 1993	Captopril	ACEi	DKD	Doubling of base-line serum creatinine concentration	[53]
RENAAL, 2001	Losartan	ARB	DKD	Doubling of serum creatinine, ESKD or death	[54]
IDNT, 2001	Irbesartan	ARB	DKD	Doubling of serum creatinine, ESKD or death	[55]
DETAIL, 2005	Telmisartan	ARB	DKD	Change in measured GFR	[56]
AMADEO, 2008	Telmisartan	ARB	DKD	Change in UACR	[57]
FIDELIO-DKD, 2020	Finerenone	NS-MRA	DKD	Kidney failure, >40% decrease in eGFR, death from kidney cause	[58]
CREDENCE, 2019	Canagliflozin	SGLT2i	DKD	ESKD, doubling of the serum creatinine level, or death from renal or cardiovascular causes	[59]
DAPA-CKD, 2020	Dapagliflozin	SGLT2i	CKD/DKD	Sustained ≥ 50% reduction in eGFR, ESKD, or death from renal or cardiovascular cause	[60]
EMPA-KIDNEY, 2023	Empagliflozin	SGLT2i	CKD/DKD	ESKD, a sustained reduction in eGFR to <10 mL/min/1.73 m^2^, renal death, or a sustained decline ≥ 40% in eGFR	[61]
FLOW, 2024	Semaglutide	GLP-1 RA	DKD	Persistent ≥ 50% reduction in eGFR, reaching ESKD, death from kidney disease or death from CV cause	[62]
ASCEND, 2010	Avosentan	ETA RA	DKD	Doubling of serum creatinine, ESKD, death	[63]
SONAR, 2019	Atrasentan	ETA RA	DKD	Doubling of serum creatinine, ESKD	[64]
ZENITH-CKD, 2023	Zibotentan + Dapaglifozin	ETA RA + SGLT2i	CKD	Change in UACR	[65]
NCT05182840, 2024	BI 690517	ASI ± SGLT2i	CKD	Change in UACR	[66]

## Data Availability

Not applicable.

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
