# Peer review of "Molecular Targets of Novel Therapeutics for Diabetic Kidney Disease: A New Era of Nephroprotection"

_ijms, 2024, doi:10.3390/ijms25073969_

Round 1

Reviewer 1 Report

Comments and Suggestions for Authors

Mazzieri et al summarise the current status of molecular targets for diabetes kidney disease and link it to potential renoprotection.

The MS needs careful grammatical assessment with countless errors. too many to summarise below.

1.     Aldosterone synthase  inhibitors As recently reported in the Lancet these need to be included in the abstract and could be linked to MRAs that they already mentioned in the table as reference 58.

2.     Diabetics people – line 31 this is not an acceptable term, consider diabetic individuals of subjects with diabetes 

3.     Finerenone -  include the FIDELITY analysis which is a pooled analysis of Figaro and Fidelio.

4.     Figure - add ASI with MRA, GLP1 Ras are not really proven to be haemodynamic, query add a? to GLP!, no evidence for SGLT2i being anti-inflammatory query add a? to SGLT2i, Consider ETA as haemodynamic 

5.     LosArtan, a correct line 250

6.     Incretin agonist - clinical The authors should mention that there are now GLP1/ glucagon agonist and Triple agonists (GLP1, GIP,glucagon agonists) and renal data are forthcoming 

7.      Incretin agonist - basic consider mentioning glucagon and the kidney with glucagon agonism in terms of the kidney now been considered (see recent paper by Scherers group in cell metabolism Wang et al 2024) and also the molecular actions of GLP1 agonism should be described (eg Sourris et al  KI 2024)

8.     reference errors example reference 53 line 611.

Comments on the Quality of English Language

See comment above, needs some work

Author Response

We thank the reviewer for her/his valuable and insightful comments. The authors have carefully considered the constructive comments and tried the best to address every one of them.

Reviewer 2 Report

Comments and Suggestions for Authors

In this well-structured review, Mazzieri et al. try to describe a new concept of nephoprotection.
The authors have achieved their aim, making a well-done review which fully describe the molecular target and therapeutic strategy of DKD.

The content is linear with the scientific literature and the references choose was opportune; Also, I was pleasantly impressed by the presented figure.
The  adjustament I require is an extension of the introduction section. In this form it is too short and generic. It would be interesting for the reader to have a description of what is known today about nephroprotection in the introductory part, with a greater extension of the conclusions through the amplification of the future implications of personalized medicine"
I invite the authors to review the introduction and extend it without repeting.

Although your article is structurally and descriptively well organized, with a full description of the molecular events underlying DKD and the associated treatment options, this manuscript lacks a fully introductive section.

In particular, I ask you to extend the description of the DKD and, above all, to integrate a description of the process of nephroprotection and personalized medicine.

After above consideration, this paper could be accepted for the pubblication in our journal.

Best regards

Author Response

(The authors gave the same response as above.)

Round 2

Reviewer 1 Report

Comments and Suggestions for Authors

Adequately revised

Comments on the Quality of English Language

Improved grammar